# Prenatal Ultrasound Suspicion of Cystic Fibrosis in a Multiethnic Population: Is Extensive *CFTR* Genotyping Needed?

**DOI:** 10.3390/genes12050670

**Published:** 2021-04-29

**Authors:** Chadia Mekki, Abdel Aissat, Véronique Mirlesse, Sophie Mayer Lacrosniere, Elsa Eche, Annick Le Floch, Sandra Whalen, Cecile Prud’Homme, Christelle Remus, Benoit Funalot, Vanina Castaigne, Pascale Fanen, Alix de Becdelièvre

**Affiliations:** 1Departement de Genetique, DMU Biologie-Pathologie, GH Mondor-Chenevier, AP-HP, F-94010 Creteil, France; chadia.mekki@aphp.fr (C.M.); abdel.aissat@inserm.fr (A.A.); annick.le-floch@aphp.fr (A.L.F.); benoit.funalot@aphp.fr (B.F.); pascale.fanen@inserm.fr (P.F.); 2INSERM, IMRB, Paris Est Creteil University, F-94010 Creteil, France; 3Service D’echographie Gynecologique et Obstetricale, GH Bichat-C Bernard, AP-HP, F-75018 Paris, France; veronique.mirlesse@hcuge.ch (V.M.); elsaeche@hotmail.fr (E.E.); 4Service Medecine Fœtale, Centre Hospitalo Universitaire de Geneve (HUG), S-1205 Geneve, Switzerland; 5Département de Gastro-Enterologie, Pneumologie, Mucoviscidose et Nutrition Pediatrique, CRCM, Université Paris 7, Hopital Robert Debre, AP-HP, F-75019 Paris, France; sophie.mayer@aphp.fr; 6Service de Genetique, Hopital Trousseau, AP-HP, F-75012 Paris, France; sandra.whalen@aphp.fr (S.W.); cecile.prudhomme@aphp.fr (C.P.); 7Service de Genetique, Centre Hospitalier Intercommunal de Creteil, F-94010 Creteil, France; christelle.remus@chicreteil.fr; 8Service d’Obstetrique et Gynecologie, Centre Hospitalier Intercommunal de Creteil, F-94010 Creteil, France; vanina.castaigne@chicreteil.fr

**Keywords:** Cystic Fibrosis (CF), *CFTR*, echogenic bowel, non-visualization of fetal gallbladder (NVFGB), prenatal, Africa, non-Caucasian, mutation

## Abstract

In families without a Cystic Fibrosis (CF) history, fetal ultrasound bowel abnormalities can unexpectedly reveal the disease. Isolated or in association, the signs can be fetal bowel hyperechogenicity, intestinal loop dilatation and non-visualization of fetal gallbladder. In these cases, search for CF transmembrane conductance regulator (*CFTR*) gene mutations is part of the recommended diagnostic practices, with a search for frequent mutations according to ethnicity, and, in case of the triad of signs, with an exhaustive study of the gene. However, the molecular diagnosis remains a challenge in populations without well-known frequent pathogenic variants. We present a multiethnic cohort of 108 pregnancies with fetal bowel abnormalities in which the parents benefited from an exhaustive study of the *CFTR* gene. We describe the new homozygous p.Cys1410* mutation in a fetus of African origin. We did not observe the most frequent p.Phe508del mutation in our cohort but evidenced variants undetected by our frequent mutations kit. Thanks to the progress of sequencing techniques and despite the difficulties of interpretation occasionally encountered, we discuss the need to carry out a comprehensive *CFTR* study in all patients in case of fetal bowel abnormalities.

## 1. Introduction

Cystic fibrosis (CF, OMIM 219700) is a life-threatening monogenic recessive disorder due to mutations in the cystic fibrosis transmembrane conductance regulator (*CFTR*) gene (OMIM*602421) [1]. Although the major cause of morbidity and mortality in patients is the progressive pulmonary disease, the condition in its classical form is also characterized by pancreatic exocrine insufficiency, meconium ileus, liver disease, congenital bilateral absence of the vas deferens (CBAVD) in males and a high sweat chloride concentration. However, the broad phenotypic spectrum associated with mutations in the *CFTR* gene ranges from severe classical CF with pancreatic insufficiency to late onset CFTR-related disorders (CFTR-RD) such as bronchiectasis or isolated male infertility by CBAVD [2]. Over 2000 *CFTR* variants have been identified so far. Whereas the ACMG variants classification is used in most diseases [3], the CF community has conserved two classifications: a functional one devoted to the choice of a genotype-based therapy and a clinical one allowing genetic counseling [4]. Clinical classification depends on the severity of the most frequently associated phenotype when the patients carry a severe mutation in *trans*: CF causing mutations are responsible for classical cystic fibrosis mainly with pancreatic in sufficiency, CFTR-RD variants retain residual function and are associated to mono-symptomatic disease or milder phenotype, benign variants have no clinical consequences and variants of uncertain clinical significance (VUS) are not yet classified [2]. More recently, intermediate variants were reclassified as variants of varying clinical consequences (VVCC) and can lead in some people to apparent good health in others to CFTR-related disorders, and only rarely to a more classical, albeit usually pancreatic sufficient form of CF itself [4].

Though major advances have been made in recent years regarding therapeutic approaches, particularly with targeted therapy for certain mutations, classical CF remains a severe disease for which prenatal diagnosis can be offered when both parents are known carriers of CF causing mutations [5]. Even in the absence of a family history of cystic fibrosis, CF can be suspected during prenatal period in the presence of ultrasound digestive abnormalities such as fetal hyperechogenic bowel (FEB) and intestinal loop dilatation, mostly during the second trimester of pregnancy. CF has been reported in 0.5–9.9% of cases with hyperechogenic bowel depending on the studies [6,7,8,9,10,11,12,13,14,15].

We previously reported a collection of 694 cases of fetal bowel abnormalities in which a sequential strategy of *CFTR* analysis had been applied, according to European recommendations [16]: As a first step, a screening for frequent mutations according to geographical origin was realized, and if one parent was positive, an exhaustive study of *CFTR* was done in the other parent. Our previous publication on a population mainly of Caucasian descent evidenced a high risk of cystic fibrosis in fetuses presenting with the triad associating FEB, intestinal loop dilatation, and NVFGB (Table 1). Therefore, we proposed to revise the molecular analysis strategy and search for rare mutations in both parents for the cases with this triad [15]. 

However, the search for frequent mutations according to geographical origin can be tricky. CF was long considered as predominant in people of Caucasian descent, with a CF asymptomatic heterozygous carrier rate of about 1/30 and with marked regional variations (http://www.genet.sickkids.on.ca/cftr, accessed on 28 April 2021) [17,18]. Subsequently, these conditions were found in all ethnicities, and there is little information about the “frequent” mutations in people originating from territories where CF is less frequent, such as Asia or Africa [5]. Indeed, available commercial frequent mutations kits poorly cover the genetic diagnosis in these vast regions. 

Molecular biology techniques have taken a huge leap forward in recent years, and genetic analyses are now much faster in routine than they were ten years ago. In our Ile de France laboratory, we have the opportunity for diverse recruitment based on the geographic origins of our patient population. Therefore, we have chosen to carry out complete *CFTR* analysis in both members of the couples in the event of prenatal ultrasound suspicion of cystic fibrosis. We aimed to determine the pertinence of this approach in a multiethnic population.

We report here on a collection of 108 cases of fetal bowel abnormalities, in which an exhaustive study of exons, exons/intron boundaries, and the search for large rearrangements was done in all parents of CF suspected fetuses during the last six years. We describe the genotype/phenotype relationship in a CF fetus with the triad of digestive signs and provide new data on *CFTR* variants according to geographical origin. Finally, we suggest performing an extensive *CFTR* study in all parents in case of fetal bowel abnormalities, regardless of their origins. 

## 2. Materials and Methods

### 2.1. Subjects

From January 2015 to December 2020, 132 fetuses were identified with fetal digestive abnormalities at routine ultrasound examination at the 2nd or 3rd trimester of pregnancy in eight French prenatal centers. Among them, 24 had other signs such as malformations, or had isolated digestive calcifications or isolated ascites. For these latter cases, *CFTR* analysis was undertaken in our laboratory however we chose to exclude them from the present study. Thus, we included 108 pregnancies (Figure 1). The digestive ultrasound signs were FEB defined as sonographic density equal to or greater than that of surrounding bone at the time of the ultrasound, intestinal loop dilatation, and NVFGB, each sign being isolated or associated with the other features.

### 2.2. Molecular Analysis

In accordance with the French legislation, the parents gave their informed consent to genetic analysis during a genetic counseling session. DNA was extracted from 204 EDTA blood samples (96 couples and 12 single mothers). Variant nomenclature is based on GenBank accession NM_000492.3 (*CFTR*) with nucleotide one being the first nucleotide of the translation initiation codon ATG. We used the HGVS nomenclature at the protein level for the exonic variants, whereas we chose to name the intronic variants by their legacy name as the most understandable nomenclature for the CF community. We performed exhaustive *CFTR* study in all patients searching for the 50 most frequent mutations by the Elucigene^®^CF-EU2v1 (Elucigene Diagnostics, Manchester, United Kingdom) commercial kit followed by sequencing of all the coding exons, their intronic flanking regions, and searching for the intronic recurrent mutation c.870-1113_1110delGAAT (1002-1113_1110delGAAT). For organizational purpose, sequencing was performed either by Sanger method (primers available on demand) or by high-throughput sequencing without any difference in the quality of the results observed. For high-throughput sequencing, the libraries were prepared with the Kapa HyperPlus kit, pooled and captured by a SeqCap EZ Choice custom capture panel (Roche Nimblegen, Madison, WI USA), and sequenced on the Illumina Miseq with the MiSeq Reagent Kit v2 2x150 bp (Illumina, San Diego, CA USA). The data were processed through two independent pipelines: MiseqReporter and the in-house SMAUG (BWA/GATK Goldstandard) pipeline. Variants were annotated by wANNOVAR and were then interpreted with various bio-informatics tools: SIFT (http://sift.jcvi.org/, accessed on 23 February 2021), PolyPhen-2 (http://genetics.bwh.harvard.edu/pph2/, accessed on 23 February 2021), MutationTaster (http://www.mutationtaster.org/, accessed on 23 February 2021), ClinVar Miner (https://clinvarminer.genetics.utah.edu/, accessed on 23 February 2021), CADD score (https://cadd.gs.washington.edu/snv, accessed on 23 February 2021), CFTR2 (https://cftr2.org/, accessed on 23 February 2021), CFTR-France (https://cftr.iurc.montp.inserm.fr/cgi-bin/home.cgi?, accessed on 23 February 2021), CYSMA (https://cftr.iurc.montp.inserm.fr/cysma/, accessed on 26 February 2021), Human Splicing Finder3.1 (HSF3.0) (http://www.umd.be/HSF/, accessed on 3 March 2021), RESCUE-ESE (http://hollywood.mit.edu/burgelab/rescue-ese/, accessed on 3 March 2021), and ESEfinder3.0 (http://rulai.cshl.edu/, accessed on 3 March 2021) [19,20,21,22,23,24,25]. We used the SALSA^®^ MLPA^®^ KIT P091-D2 CFTR (MRC-Holland, Amsterdam, Netherlands) to search for large rearrangements.

## 3. Results

### 3.1. Fetuses’ Phenotypes and Parental CFTR Genotypes

Among the 108 pregnancies, the initial sign was isolated FEB in more than half of the cases (70 cases, 65%) (Figure 2). In three cases, the hyperechogenic bowel was associated with ascites or digestive calcifications. Twenty-five cases (23%) had intestinal loop dilatation, with associated hyperechogenic bowel in about half of the cases (13 cases, 12% of fetuses). Two fetuses with images evocative of meconium peritonitis were included in this subgroup of 13 cases, as they were described in the ultrasound report with FEB and moderate loop dilatation. NVFGB was reported in 16 fetuses: isolated in eight cases, which was associated with intestinal loop dilatation in two cases, and with hyperechogenic bowel in five cases. The triad of signs hyperechogenic bowel, moderate intestinal loop dilation, and non-visualization of the gallbladder was seen only once in our cohort, representing the only fetus who was diagnosed with cystic fibrosis in our study (Figure 2).

Among the 108 fetuses, only one had cystic fibrosis (less than 1%). It was the fourth pregnancy of a consanguineous couple of Senegalese origin. Their first child, born in 2001, died at her first day of life in the context of meconial peritonitis. Although fetopathological analysis showed a digestive pattern compatible with CF, this potential diagnosis was not further investigated in this first child or her non-Caucasian parents. Then the couple then had two healthy children. In 2016, at 22 weeks of a fourth pregnancy, the ultrasound examination revealed the triad of signs in the fetus: hyperechogenic bowel, moderate intestinal loop dilation and non-visualization of the gallbladder. In the context of consanguinity and of the triad of fetal digestive ultrasound signs, in-depth analysis seemed recommended. An exhaustive *CFTR* analysis revealed the heterozygous novel nonsense c.4230C>A, p.Cys1410* mutation in both parents. This result made it possible to provide the couple and their family with appropriate genetic counseling. The couple chose to pursue the pregnancy without testing the fetus by an invasive procedure. However, thanks to this expected CF diagnosis, the mother was transferred to a specialized maternity for delivery. The baby girl presented at birth with meconium peritonitis due to an ileal atresia with segmental volvulus and meconial cyst requiring immediate ileocecal resection and ileostomy. The intestinal continuity was restored after 6 months. Despite a false negative result for newborn screening, the CF diagnosis was confirmed by a positive sweat test and the presence of the mutation in homozygosity. She developed a neonatal exocrine pancreatic insufficiency. *Pseudomonas aeruginosa* was first isolated at the age of seven months. Currently, at the age of four years and a half, she has an acceptable nutritional status (BMI Z-score 0.7).

Fewer than 2% of fetuses were at risk of CFTR-RD genotype according to molecular analysis in our cohort. One couple had a 50% risk of having a baby with a CFTR-RD genotype: the father was heterozygous for the p.Phe508Cys variant, and the mother was heterozygous for the three variants TG11T5, TG12T5, and p.Val562Ile. The fetus presented with hyperechogenic bowel and dilated intestinal loops. As the mother was not heterozygous for a CF mutation, no segregation study was required in this case. Based on the knowledge of complex alleles, she probably had the genotype c.[1210-34TG[12]T[5 ]](;)[1210-34TG[11]T[5];1684G>A] and was considered a carrier for two CFTR-RD variants. Another couple exhibited a 25% risk of having a baby with a CFTR-RD genotype, as the father and the mother were heterozygous for p.Leu997Phe and p.Met348Lys respectively. The fetus had isolated hyperechogenic bowels.

The CFTR-RD genotype is not eligible for prenatal diagnosis in France therefore, the at-risk fetuses were not tested. Thus, we cannot conclude whether these two phenotypes were due to a CFTR default. Except for the CF fetus, we did not encounter any other situation where both parents were carriers of VVCC/VUS/CF- or CFTR-RD variants.

### 3.2. CFTR Genotypes in Parents from Different Ethnic Groups

Among the 204 parents, at least one *CFTR* variant was identified in 32 individuals (16%), ranging from CF mutation to CFTR-RD variant with low penetrance and VUS (Table 2).

VUS were very rare variants for which the benign or pathogenic effect was unclear in the CFTR2, CFTR-France, and ClinVar Miner databases. We evaluated their pathogenicity with different in silico tools and according to their frequency (Table 3). Indeed, we assumed that variants with a frequency similar or higher than that of the frequent CF mutation N1303K (0.018% allelic frequency in gnomAD) were probably non CF-causing. The variants observed were a CF mutation in four parents (12% of the identified variants), a VVCC mutation in six parents (17%), and a CFTR-RD variant in 15 parents, including 12 classified as low penetrant variants, and a VUS in nine cases (Figure 3). Among the VUS, five were considered as non CF-causing (15% of the variants), whereas interpretation remained complicated for four (12% of the variants) for which we could not conclude to a severe or benign effect. The exhaustive search for a CF or VVCC mutation eligible to prenatal diagnosis was negative in the spouses of the VUS carriers.

Among the three CF mutations (in four parents), only one (3849+10kbC>T) was identified by the Elucigene^®^CF-EU2v1 kit, which detects the 50 most frequent *CFTR* mutations. The p.Cys1410* and p.Arg764* mutations were detected by exhaustive *CFTR* analysis. Regarding the five different variants classified as VVCC (in six parents), two were completely identified by the Elucigene^®^CF-EU2v1 kit (p.Asp1152His and TG13T5), one was incompletely detected (TG12T5 very probably in complex allele with the undetected p.Ser977Phe variant), and two were missed by the frequent mutations kit (p.Pro750Leu and p.Gln237Glu).

Although the geographical origin was not available for 24 individuals (12 couples), we observed a very mixed population in the 183 other parents. Less than half of our recruitment (34%) was of Caucasian descent: forty-five individuals were from metropolitan France (22%) and 24 (12%) were of other Caucasian origin including 12 (5.5%) of Iberian origin. Among the other ethnic groups, 35 individuals (16%) were from Northern Africa (Algeria, Morocco, and Tunisia), 40 (19%) were from Sub-Saharan Africa (mainly Mali, Senegal, Cameroun, and Congo), and 16 (8%) were of another non-Caucasian region (mixed origins such as Reunion Island, Mauritius, Guadeloupe) (Figure 4).

We focused on the variants in the different ethnic populations (Table 2). In the 45 individuals of French origin, at least one rare variant was found in five individuals. Only two CF/VVCC were identified, one of them being detected by the frequent mutation kit. Only one VUS for which the severe effect could not been ruled out was detected (less than 1% alleles).

In the 35 individuals from Northern Africa, the p.Asp1152His VVCC was reported twice. No other rare pathogenic variant associated with CF was detected. The CFTR-RD p.Leu997Phe variant was found in three parents. It was also carried out by one parent from European and Northern African descent in the “other origins group”.

Among the 40 individuals from Sub-Saharan Africa, seven were heterozygous for a rare variant, including three CF mutations (p.Arg764* in one and p.Cys1410* in two parents) and two VUS for which a possible CF effect could not be ruled out (p.Glu282Asp and p.Gly1173Ser). None of the six variants identified was detected by the frequent mutation kit. Moreover, if the two stop gain mutations were easily classified as CF-causing, interpretation was uncomfortable for other variants.

In the 20 individuals originating from Asia, only one had a rare variant, the TG13T5 allele, at the heterozygous state.

A parent from Guadeloupe carried the p.Pro750Leu VVCC variant. The c.870-1095A>C variant was found in a parent of Mauritius, an island with a mixed population. This deep intronic variant was uneasy to interpret. It is located in intron 7, not far from the recurrent intronic c.870-1113_1110del pathogenic variant, which is a known VVCC that was identified by abnormal mRNA splicing, whereas in silico tools do not predict huge splice changes.

## 4. Discussion

### 4.1. Ultrasound Digestive Abnormalities

Based on our previous study, we only considered in the present study three signs evocative of cystic fibrosis: fetal hyperechogenic bowel (FEB), intestinal loop dilatation and non-visualization of fetal gallbladder (NVFGB). Ascites or digestive calcifications are not strong predictive features and were taken into account only if they had at least one of the associated signs above [15].

Indeed, isolated fetal ascites has been very rarely reported in CF. However, a recent publication mentioned ascites as a warning sign for CF [26]. In the reported case, after an uneventful pregnancy, fetal ascites was observed at 32 weeks and the child was born at 34 weeks. Abdominal ultrasound showed dilated bowel loops in the neonate. The child also displayed intestinal malrotation and mesenteric hernia and was homozygous for the p.Phe508del mutation.

Hyperechogenic bowel is an aspecific sign, as it can be observed in cases with chromosomal abnormalities, congenital infections, intestinal abnormalities, intrauterine growth restriction, intra-amniotic bleeding and many times in disease-free condition [14,27]. Although CF can be diagnosed in fetuses with isolated FEB, this relatively common sign becomes more specific when associated with other features such as intestinal loop dilatation or non-visualization of the gallbladder [15,28]. Fetal dilated bowel loops are a sign of intestinal obstruction of various etiologies such as bowel atresia, intestinal volvulus, and meconium ileus. However, it also can be transient or found in fetuses with no abnormality at postnatal evaluation [29].

NVFGB is a rare condition, with an estimated incidence of 0.1% pregnancies [30]. Among the 16 fetuses with gallbladder abnormalities in our cohort, eight presented with isolated NVFGB. None of these eight fetuses had a CF or a CFTR-RD genotype. A recent publication of 18 cases of NVFGB showed that in about 70% of cases, gallbladder was visualized later in pregnancy or at birth as was confirmed by a meta-analysis reporting on 273 cases [31]. According to this meta-analysis, the benign condition of gallbladder agenesis was observed in 25% of cases with previous NVFGB, and a severe condition was found in about 10% cases (1.9% chromosomal abnormalities, 4.8% biliary atresia and 3.1% CF cases). In the cohort of Duguéperoux et al., reporting on 37 fetuses referred for NVFGB, FEB was associated in the five CF-affected fetuses [28]. However, a recent study including 37 CF fetuses showed isolated NVFGB in five (13.5%) [32]. Thus, although we did not find any *CFTR* deleterious variant in our group, CF must be considered in case of NVFGB, isolated or with other digestive signs, and molecular analysis must be proposed to the parents.

Remarkably, the only CF fetus in our study presented with the triad of signs for which we evaluated the highest risk of Cystic Fibrosis previously [15]. This triad of signs is very rare in our experience: in our previous study, we observed this triad in eight fetuses (1.15%) among the 694 cases referred for ultrasound digestive abnormalities [15], and in the present study we observed it only once among 132 fetuses.

### 4.2. Low Risk of CF in Our Cohort of Diverse Geographical Origins

The risk of CF diagnosed on ultrasound digestive abnormalities ranges from 0.5 to 9% depending on the studies [13,14]. The frequency of CF in our cohort was very low, less than 1%. Notably, it was lower than the latest cohort we published in our group, recording 694 pregnancies, in which we had evaluated the global CF risk at about 2.15% [15]. Our much smaller number of CF cases, while we were conducting a comprehensive *CFTR* study, could be due to our multiethnic population and to a less stringent selection of patients, in a willingness of practitioners to take as little risk as possible at a time when massive sequencing has become more affordable. Moreover, systematic neonatal CF diagnosis was implemented in France since 2002, detecting the frequent severe mutations in CF neonates but also incidentally healthy heterozygous carriers [33]. This, added to the recent legal obligation to inform relatives in the case of severe mutations identified in our country, has the consequence of increasing the cascade screening of relatives and frequent mutation searches in spouses of heterozygotes prior to pregnancy.

Moreover, the majority of patients are of Caucasian descent in most studies. In this population, CF is a relatively frequent genetic disease with a 1/3000 prevalence and a carrier frequency for a CF mutation estimated at about 1/30 [17,18]. Previous studies evidenced frequent mutations in the parents of fetuses with digestive abnormalities [9,10,15,27]. Independently of the CF cases, we identified in our previous study a 2.8% CF carrier frequency in a cohort of 465 fetuses, and recently, Ronin et al. reported that one of the parents was heterozygous for a frequent mutation in 2.9% (of 325 pregnancies) which is similar to the general population [15,27]. We present here a multiethnic cohort with only 34% of patients of Caucasian origin. This could explain why we did not observe the most common p.Phe508del mutation in our population.

Cystic fibrosis frequency is very poorly known in territories such as Asia or Africa. The disease has a 1/15,100 incidence in the African American population [34], but its frequency in the Sub-Saharan population remains unknown [35,36]. Similarly, CF birth prevalence in the Asian American population was estimated at 1/35,100 [34]. Few studies have evaluated the CF incidence in Asia, but there are huge variations among the countries, from 1/2985 live births in Jordan (similar to that in Caucasian population) to 1/350,000 live births in Japan [37]. In many regions, the disease is most probably under-diagnosed: physicians do not consider it, the phenotype being very similar to other disease such as protein energy malnutrition, diarrheal diseases, tuberculosis, chronic pulmonary infections or HIV/AIDS. Moreover, the diagnosis is difficult to establish, since sweat test and genetic testing are often inaccessible [35].

### 4.3. The CF Frequent Mutation Kits Are Not Appropriate in Non-Caucasian Populations

The only frequent pathogenic variants that we observed in our cohort were the VVCC TG13T5 and p.Asp1152His. The latter is evidenced by the Elucigene^®^CF-EU2v1 kit, which detects 50 variants, but not by the CF30v2 Elucigene kit usually dedicated to neonatal screening in France. None of our patients was a carrier for the very frequent p.Phe508del mutation.

In our cohort, 36% of patients were from Northern or Sub-Saharan Africa. We found the p.Asp1152His VVCC and the p.Leu997Phe CFTR-RD variants in patients from Northern Africa. We did not observe neither the p.Glu1104* mutation frequent in Algeria, Tunisia, and Libya [38,39], nor the 3120+1G>A mutation, which accounts for 53% of African-American *CFTR* mutations and was also found in Afro Cubans, Cameroonians, Rwandans, Zambians, and South Africans and is included in the Elucigene^®^CF-EU2v1 kit [35,38]. We also did not detect the recurrent c.3469-1304C>G CF mutation recently identified in patients of African origin [40], which was searched for in 13 African parents. Instead, we identified the rare p.Arg764* and the private p.Cys1410* mutations, which are not included in the Elucigene^®^CF-EU2v1 kit. The p.Arg764* mutation was reported in ClinVar, in six CF patients in the CFTR-France database and in 30 CF patients in the CFTR2 database [24,25]. It was first described in a woman of Afro-American origin (http://www.genet.sickkids.on.ca/, 28 February 2021) but is not specific of African population, since it was also observed in Europeans [41,42]. The p.Cys1410* mutation was never described in patients. According to the rules of NMD, this stop mutation should not elicit nonsense-mediated mRNA decay (NMD) [43]. The predicted resulting mutated CFTR protein would be deleted of its 70 COOH-terminal amino acids, including the PDZ domain. The cytoplasmic C-terminal domain of CFTR is known to be involved in apical localization, protein–protein interactions through the PDZ binding domain and the maintenance of the mature protein. Indeed, although this mutation has never been reported in humans, a previous CFTR systematic C-terminal truncation study showed a dramatic decrease of CFTR mature protein [44].

There is a well-known bias in the CF frequent mutation kits, as they were designed for Caucasian populations. As described in Hughes et al., in a retrospective study of CF infants born in New York City, the ACMG-23 kit detected 81.8% of mutated alleles in White patients, 59.9% in Hispanic patients, 62.5% in Asian patients (only 4 patients), and only 50% in Black American patients [45]. The Illumina 139-VA kit increased the detection rates to 91%, 71.1%, 75%, and 68.2% respectively. In their study about American African diaspora, Stewart and Pepper reported that the p.Phe508del mutation was the most frequent in this population (29.4%), although it was lower than in the Caucasian population. The mutation remained unknown in 56.5% [38]. While the disparity in identified mutations in non-Caucasian populations versus Caucasian populations has been known for a long time, Nappo et al. recently objectified it thanks to the numerous information of the Genome Aggregation Database (gnomAD) [46]. Among a selection of 222 CF or CFTR-RD causing variants, 79% were present in the European non-Finnish population, 23% were present in the African population, and 25% were present in the Asian population (19% South Asian, 6% East Asian). This study revealed 12 *CFTR* variants specific of the African population and 11 of the South Asian population. However, in the gnomAD database, populations are mixed, the African population comprising Afro-Americans, North Africans, and Black Africans. Asia is divided into East Asia and South Asia, but theses territories remain very large with population specificities inside.

In our previous study of 694 fetuses, only two CF fetuses were carriers of two rare CF mutations (0.2% of cases). One homozygous fetus was from a Turkish consanguineous family, whereas the second was of Caucasian origin and had two different mutations [15]. In order to minimize the risk of CF for couples of geographical origins in which CFTR variance is understudied or poorly understood, performing an extensive study of the *CFTR* gene is appropriate. In the interest of health equality, it could be legitimate to propose an extensive study to all couples whatever their origin is to obtain a similar low risk of CF in the fetus.

### 4.4. Difficulties in Interpreting Rare Variants

Since the discovery of the *CFTR* gene in 1989, technological advances have made it possible to analyze the gene ever better. The technological progress must take place in the medical context. Current recommendations for *CFTR* analysis, in the context of spousal analysis of heterozygotes or ultrasound suspicion of Cystic Fibrosis, are to look for the most frequent mutations according to geographical origin. However, this is not adapted to populations originated from very large territories for which the mutations are unknown.

However, making an extensive *CFTR* study means also identifying rare variants of which the interpretation can be difficult. In our study, 27% of the rare variants identified were considered as variants of uncertain clinical significance (VUS). Thanks to their frequency in the gnomAD database and/or bio-informatics prediction tools, we could eliminate a severe CF effect for 55% of them. Fortunately, we did not encounter any case in our cohort of a couple with one member carrying a VUS and the other member carrying a mutation or a VUS. According to the ACMG guidelines, VUS should not be used in clinical decision-making. However, it is recommended to update regularly if the knowledge has changed on any VUS [3]. In such cases, the measurement of fetal digestive enzymes in amniotic fluid can be helpful. However, this invasive procedure must be performed before 22 weeks of gestation to be valuable, and digestive abnormalities are mainly observed after 22 weeks [47]. VUS interpretation is especially complex in case of ultrasound bowel abnormalities because of the lack of specificity of the fetal signs: the identified *CFTR* variants are not necessarily related to CF. In addition, we have previously shown that fetal digestive abnormalities can also lead to the detection of CFTR-RD genotypes and in most cases of non-pathogenic *CFTR* genotypes [15]. Regarding the CFTR-RD risk, the question of the knowledge of the genotype arises: how to decide the benefit/risk balance between the very high sensitivity of the extensive study and the risk of generating undesired genetic data, which may cause anxiety to the patient’s families?

### 4.5. The Role of the Laboratory

In the current context of massive sequencing, the analytic step for extensive *CFTR* analysis is accessible to many laboratories. However, although bioinformatics tools and databases are developed as assistance to the interpretation, a good knowledge of variants, their penetrance, and the known complex alleles is mandatory to provide a correct molecular diagnosis [24,25]. This expertise is particularly important in case of very rare or unknown variants, for which clinical observations are inconclusive, and in the context of ultrasound suspicions where the phenotype is not specific. However, we are fortunate in France to benefit from a very active network of laboratories (GenMucoFrance), so that the interpretation of rare variants is collegial.

### 4.6. The Role of the Geneticist

The clinical geneticist has to answer several questions, among which are the indications for prenatal testing and implications of the results. If the proposal of prenatal diagnosis seems unequivocal for severe CF pathogenic variants, it is much more debated for VVCC pathogenic variants associated to moderate late onset CF or to CFTR-RD. The geneticist must also determine if there is an indication for familial analysis.

Prenatal diagnosis in France is strictly supervised and only available for severe and untreatable diseases. It can lead to termination of pregnancy but also to a better neonatal care and follow up of the newborn child. A multidisciplinary team including a geneticist, an obstetrician, a psychologist, a pediatrician, and a specialist of the considered disease evaluates each case of prenatal diagnosis. This process allows each case to be discussed according to the current state of knowledge and to support the couples to limit the psychological impact, whether they choose termination of pregnancy or not. This choice can be particularly difficult for the couple in late pregnancies and with no familial history of CF. The development of targeted therapies and the incredible breakthrough of Elexacaftor–Tezacaftor–Ivacaftor for the benefit of the patients bearing at least one p.Phe508del allele will undoubtedly change our handling of these pregnancies [48].

## Figures and Tables

**Figure 1 genes-12-00670-f001:**
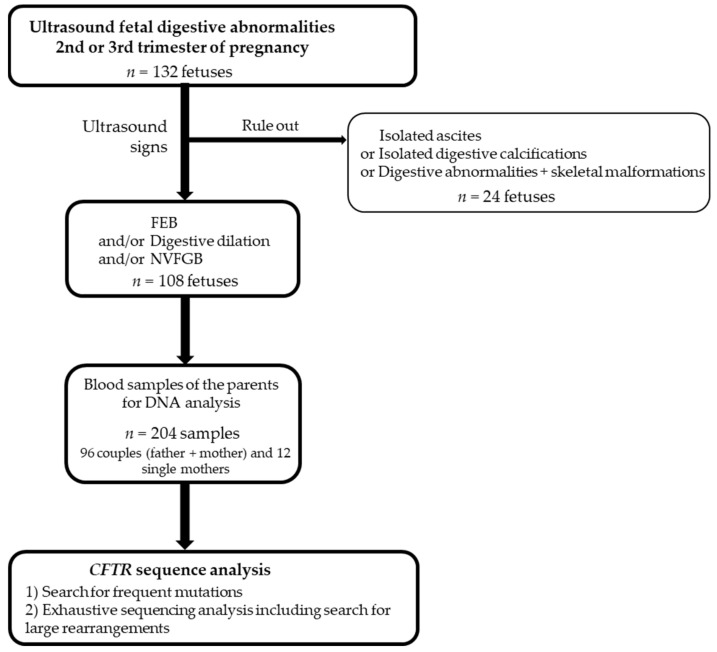
Flow diagram of inclusion criteria and molecular analyses. FEB: fetal echogenic bowel, NVFGB: non-visualization of the fetal gallbladder.

**Figure 2 genes-12-00670-f002:**
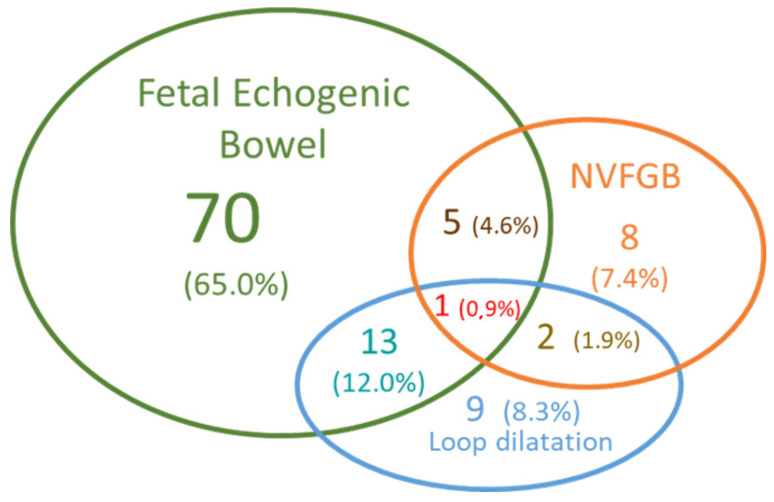
Venn diagram representing the different signs observed in 108 fetuses. We considered only the cases with FEB (fetal echogenic bowel) (green circle), intestinal loop dilatation (blue circle) or/and NVFGB (non-visualization of fetal gallbladder) (orange circle). The number of fetuses is indicated in each circle was followed by the percentage among the 108 fetuses.

**Figure 3 genes-12-00670-f003:**
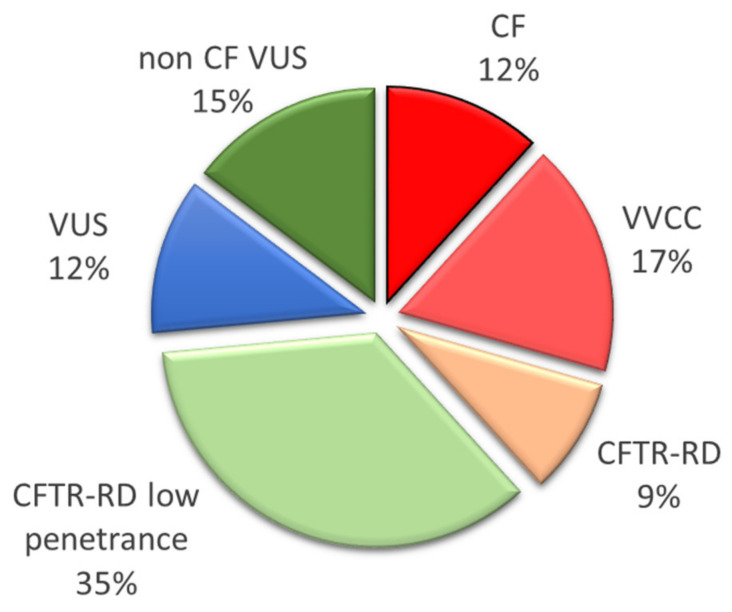
Repartition of the variants alleles identified in the fetuses’ parents regarding their pathogenicity in the CF context. CF, variants causing Cystic Fibrosis disease; VVCC, variants of varying clinical consequences; CFTR-RD, variants causing CFTR-related disorder; VUS, variants of uncertain significance; non-CF VUS, variants of uncertain significance non causing cystic fibrosis disease.

**Figure 4 genes-12-00670-f004:**
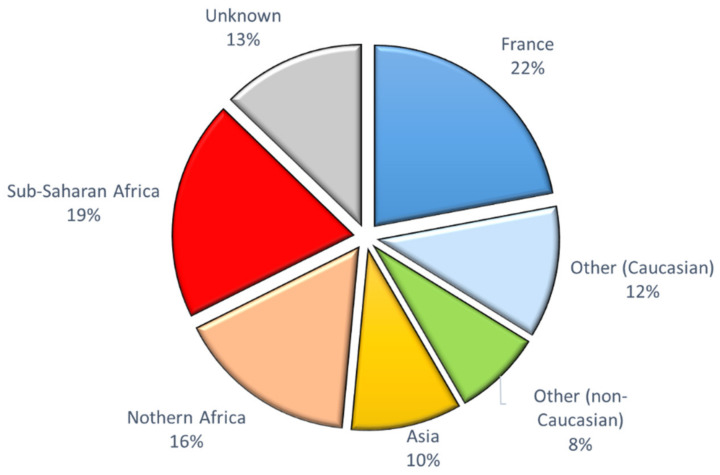
Prevalence of geographical origin of the fetuses’ parents (*n* = 204).

**Table 1 genes-12-00670-t001:** The value of the ultrasound signs for the prenatal CF diagnosis (adapted from de Becdelievre et al. [15]).

Ultrasound Signs	Likelihood Ratio (LR)	Number of Fetuses Observed with the Ultrasound Abnormalities in [15]	
		CF Fetuses	Non-CF Fetuses
**Isolated signs**			
Abdominal calcifications	0	0	12
Isolated NVFGB	0	0	6
Ascites	**1.19**	0	14
FEB	0.42	8	376
Loop dilatation	**3.10**	3	26
**Associations of signs**			
Meconium peritonitis *	**1.94**	2	26
FEB + loop dilatation	**2.65**	5	42
FEB + NVFGB	**3.87**	2	14
Triad: FEB + loop dilatation + NVFGB	**31.40**	4	4

In this study, the fetuses were mainly of Caucasian origin. The likelihood ratio (LR) considers the sensitivity and specificity of the signs and quantifies the value of the patterns for the diagnosis of CF. LR>1 (in bold) indicates that a CF fetus has an increased risk of presenting a given pattern as compared with a non-CF fetus. FEB: Fetal echogenic bowel, NVFGB: non-visualization of fetal gallbladder. * Meconium peritonitis ultrasound signs combine hyperechogenic areas with peritoneal calcifications with or without dilated bowel loops, ascites and meconium pseudocyst.

**Table 2 genes-12-00670-t002:** *CFTR* variants identified in parents with different geographical ethnicity.

Origin	cDNA Name	Protein Name	Legacy Name	dbSNP	Nb of Carriers	Classification in CFTR2	Classification in CFTR-France	ClinVar Miner	Our Interpretation
**France**									
	c.[**3718-2477C>T**]; [709C>G]	p.[?];[Gln237Glu]	3849+10kbC>T/Q237E	rs75039782 rs397508784	1	CF (PS); VVCC	VVCC; VVCC	P (1), VUS (1), Lik. B (1), B (1)/ VUS (1)	CF/VVCC, genotype compatible with mild CF
	**c.1210-34TG[11]T[5]**	p.?	TG11T5	no rs	2	VVCC	VUS	VUS (1)	CFTR-RD with low penetrance
	c.2900T>C	p.Leu967Ser	L967S	rs1800110	1	VVCC	VUS	P (1), Lik. P (1), VUS (8)	VUS non CF
	c.3590A>G	p.His1197Arg	H1197R	no rs	1	not reported	not reported	not reported	VUS
	c.224G>A	p.Arg75Gln	R75Q	rs1800076	2	non CF	non disease-causing	P (1), Lik. B (5), B (6), VUS (4),	CFTR-RD with low penetrance
**Northern Africa**									
	**c.3454G>C**	p.Asp1152His	D1152H	rs75541969	2	VVCC	VVCC	P (15), Lik. P (1)	VVCC
	c.2991G>C	p.Leu997Phe	L997F	rs1800111	2	non CF	CFTR-RD	not reported	CFTR-RD
	^(*)^c.[**1210-34TG[12]T[5]**](;) [**1210-34TG[11]T[5];**1684G>A]	p.[?](;)[?;Val562Ile]	TG12T5/ TG11T5/V562I	rs1800097	1	VVCC; VVCC; non CF	CFTR-RD;VUS;VUS	P (1), Lik. B (1), VUS (6)	CFTR-RD
	c.2002C>T	p.Arg668Cys	R668C	rs1800100	1	non CF	CFTR-RD	Lik. B (1), B (3), VUS (11)	CFTR-RD, low penetrance
	^(*)^c.2991G>C/**c.1210-34TG[12]T[5]**/c.2930C>T	p.Leu997Phe/p.?/p.Ser977Phe	L997F/TG12T5/S977F	rs1800111 rs141033578	1	non CF; VVCC	CFTR-RD; VVCC	not reported/VUS (1), P (2), Lik. P (1)	VVCC/CFTR-RD low penetrance
**Sub-Saharan Africa**									
	c.4230C>A	p.Cys1410*	C1410X	no rs	2	not reported	not reported	not reported	CF
	c.2290C>T	p.Arg764*	R764X	rs121908810	1	CF (PI)	CF	P (7)	CF
	c.846A>T	p.Glu282Asp	E282D	rs142864834	1	not reported	not reported	VUS (4)	VUS
	c.4333G>A	p.Asp1445Asn	D1445N	rs148783445	1	not reported	VUS	Lik. P (1), Lik. B (1) VUS (9)	VUS, non CF
	c.3517G>A	p.Gly1173Ser	G1173S	rs368393738	1	not reported	not reported	VUS (2)	VUS
	c.4243-20A>G	p.?	NA	rs138025486	1	not reported	VUS	Lik. B (3), VUS (3)	VUS, non CF
**Other geographical origins (Asia, Europe, mixed origins, unknown)**									
**Sri Lanka**	**c.1210-34TG[13]T[5]**	p.?	TG13T5	no rs	1	VVCC	VVCC	not reported	VVCC
**Germany**	c.[220C>T;890G>A]	p.Arg74Trp/p.Arg297Gln	R74W/R297Q	rs115545701 rs143486492	1	not reported	VUS	Lik. P (4), Lik. B (2), B (1), VUS (5)/ Lik. B (6), B (1), VUS (2)	VUS non CF
**Italy**	c.1043T>A	p.Met348Lys	M348K	rs142920240	1	not reported	VUS	Lik. B (5), B (1), VUS (3),	VUS non CF
**Guadeloupe**	c.2249C>T	p.Pro750Leu	P750L	rs140455771	1	VVCC	CFTR-RD	P (2), Lik. P (5), VUS (5)	VVCC
	c.853A>T	p.Ile285Phe	I285F	rs151073129	1	not reported	not reported	Lik. B (2), B (1)	VUS non CF
**Reunion Island**	c.224G>A	p.Arg75Gln	R75Q	rs1800076	1	non CF	non disease-causing	P (1), Lik. B (5), B (6), VUS (4),	CFTR-RD, low penetrance
**Mauritius**	c.870-1095A>C	p.?		rs540046342	1	not reported	not reported	not reported	VUS
**Caucase/Northern Africa**	c.2991G>C	p.Leu997Phe	L997F	rs1800111	1	non CF	CFTR-RD	not reported	CFTR-RD low penetrance
**Unknown**	c.1523T>G	p.Phe508Cys	F508C	rs74571530	1	non CF	CFTR-RD	P (2), Lik. B (1), B (5), VUS (3),	CFTR-RD low penetrance
	c.3705T>G	p.Ser1235Arg	S1235R	rs34911792	1	non CF	VUS	Lik. B (5), B (7), VUS (3)	CFTR-RD low penetrance
	**c.1210-34TG[12]T[5]**	p.?	TG12T5	no rs	1	VVCC	CFTR-RD	not reported	CFTR-RD

The patients were grouped according to their geographical origin (grey background). Abbreviations: NA: Not applicable; CF: Cystic Fibrosis, CFTR-RD: CFTR-related disease, VVCC: Variants of varying clinical consequences, VUS: Variant of uncertain significance, P: Pathogenic, Lik. P: Likely pathogenic, Lik. B: Likely benign, B: Benign. For the CF-causing variant, the status PS/PI (pancreatic sufficiency/insufficiency) from the CFTR2 database is indicated. In the cDNA name column, the variants in bold are identified by the frequent mutation Elucigene^®^CF-EU2v1 kit. In the ClinVar Miner column, the numbers of reported observations in the database are in brackets The items related to “drug response”, “not provided”, and “risk factor” were excluded, as well as the items related to CFTR-France in the ClinVar Miner counts. (*): probably in a complex allele according to the frequency of these complex alleles in databases, but no family co-segregation study was realized. One asymptomatic mother was compound heterozygous for two pathogenic variants with the 3849+10kbC>T/p.Gln237Glu genotype, which was compatible with mild CF.

**Table 3 genes-12-00670-t003:** In silico prediction for the variants interpreted as VUS in our laboratory.

Variant	gnomAD Frequency (European Non-Finnish)	Structural Effects Predicted by CYSMA	SIFT/PolyPhen-2	Mutation Taster	CADD Score	In Silico Splicing Analysis Tools
p.Leu967Ser	0.10%	Solvent accessibility: buried No steric clashes	T/D	Disease causing	24.3	NR
p.His1197Arg	0.00090%	Solvent accessibility: exposed No steric clashes	T/B	Polymorphism	12.3	NR
p.Glu282Asp	0.000%	Solvent accessibility: buried No steric clashes	T/D	Disease causing	24.5	NR
p.Asp1445Asn	0.029%	Solvent accessibility: buried No steric clashes	T/D	Disease causing	26.3	NR
p.Gly1173Ser	0.000%	Solvent accessibility: wild-type G1173 exposed, mutant G1173S buried Steric clashes	T/B	Polymorphism	2.1	NR
c.4243-20A>G	0.0016%	Not predicted	Not predicted	Not predicted	14.9	*HSF*: Putative branch point disappearance, but strong branch points in the vicinity of the affected one → putative low impact on splicing
p.Arg74Trp	0.033%	R74 is involved in the CFTR pore construction Solvent accessibility: wild-type R74 exposed, mutant R74W buried Steric clashes	T/D	Disease causing	25.3	NR
p.Arg297Gln	0.12%	Solvent accessibility: wild-type R297 buried, mutant R297Q exposed No steric clashes	T/B	Disease causing	24.8	NR
p.Met348Lys	0.021%	Solvent accessibility: wild-type M348 buried, mutant M348K exposed Steric clashes	D/D	Disease causing	27.3	*HSF*: Putative acceptor site creation (weaker than the natural site)
p.Ile285Phe	0.000% *	Solvent accessibility: buried Steric clashes	D/D	Disease causing	27.8	NR
c.870-1095A>C	0.013%	Not predicted	Not predicted	Not predicted	5.0	*ESEfinder3.0*: Deep Intronic SC35 site creation → putative low impact on splicing

VUS: Variant of uncertain significance. The gnomAD frequencies of the VUS were compared to that of the frequent mutation p.Asn1303Lys: 0.018%.* The frequency of the p.Ile285Phe variant is below 0.018%, but one homozygous individual is reported in gnomAD. CYSMA: Cystic Fibrosis Missense Analysis (https://cftr.iurc.montp.inserm.fr/cysma/, accessed on 26 February 2021); SIFT: Sorting Intolerant From Tolerant (http://sift.jcvi.org/, accessed on 23 February 2021) (T = Tolerated, D = Damaging); PolyPhen-2 (http://genetics.bwh.harvard.edu/pph2/, accessed on 23 February 2021) (B = Benign, D = Damaging); MutationTaster (http://www.mutationtaster.org/, accessed on 23 February 2021). CADD: Combined Annotation Dependent Depletion (https://cadd.gs.washington.edu/snv, accessed on 23 February 2021) (we estimated as likely pathogenic the variants with a score >20). Splicing analysis tools: for exonic variants, only the creation of cryptic acceptor/donnor sites or suppression of natural acceptor/donor sites were searched for. NR = Non-Relevant. For intronic variant, only results with tools showing a change have been reported in the table: Human Splicing Finder3.1 (HSF3.0) (http://www.umd.be/HSF/, accessed on 3 March 2021), RESCUE-ESE (http://hollywood.mit.edu/burgelab/rescue-ese/, accessed on 3 March 2021); ESEfinder3.0 (http://rulai.cshl.edu/, accessed on 3 March 2021).

## Data Availability

Not applicable.

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
