# Peer review of "Prenatal Ultrasound Suspicion of Cystic Fibrosis in a Multiethnic Population: Is Extensive CFTR Genotyping Needed?"

_genes, 2021, doi:10.3390/genes12050670_

Round 1

Reviewer 1 Report

Major comment

  1. Please consider grammatical review and editing from someone whose primary language is English (which will help with some of the minor comments below).
  2. Page 3, subjects section: For readers to understand the scope of work done at the hospital, consider inclusion of a consort diagram starting with the total number of prenatal ultrasounds done during the period of time.
  3. In the intro, from the author’s previous work, include the sensitivity and specificity of the prenatal ultrasound observed “triad” for diagnosis of CF. Would also consider a table describing the sensitivity and specificity for diagnosis of CF using a single one of the diseases listed in the triad, versus 2 of 3 versus all three. Such a table could help a clinician decide on the utility of suggesting CFTR sequencing for the parents based on both the signs on prenatal ultrasound and other factors (e.g. family history, previous history of miscarriages, etc).
  4. Section 3.2 Please reference to which databases the authors are referring (line 204) and be more specific about the in silico tools being utilized to assess pathogenicity (fine to reference legend for figure 2).
  5. The discussion could be shortened as there is some redundancy.

Minor comments

  1. Page 2, lines 54-55, please clarify what the phrase “conserved a functional and a clinical variants classification”
  2. Page 2, lines 55-57 it seems that the authors are stating that clinical classification depends on the severity of the genotype dictated phenotype (although there is clearly variability in that correlation as demonstrated by the work done by Mike Knowles in people homozygous for F508del). The list that follows is a bit confusing—it needs to be edited, referenced and the terms defined.
  3. Page 2 line 80, change “are” to “is”
  4. Define what is meant by the phrase “severe CF mutations” with first use.
  5. Page 2, line 86, replace large recruitment with something like “we have the opportunity for diverse recruitment based on the geographic origins of our patient population”. Would also recommend breaking that very long sentence into two sentences.
  6. Page 2. Line 94: do the authors mean in all patients presenting with bowel abnormalities on prenatal ultrasound, or all patients presenting with signs and symptoms of CF?
  7. Page 3, line 102, change “retained” to “included”.
  8. Page 3, lines 102-105, FEB and NVFGB were defined in the intro, so don’t need to be spelled out again here.
  9. Page 3, lines 118-119: Please explain what is meant by “Depending on the relative emergency of the molecular diagnosis,” e.g. under what circumstances was the diagnosis considered an emergency and why?
  10. Page 4, line 159 change “on” to “of” or “born to”
  11. Page 4, line 174: change “satisfying” to “acceptable”.
  12. Page 4, line 181, change “on” to “of”
  13. Page 4, lines 185-186, please clarify the points of the sentence starting with CFTR-RD genotype. Do the authors mean “amenable” rather than “eligible”? Neither parent was tested or the pregnancy was multiple?
  14. Page 11, lines 320-321, do the authors mean other fetuses reported in the literature or in their previous case study or?
  15. Page 11 line 345-add references after abnormalities.
  16. Page 11, line 355, add “the” before Asian.
  17. Page 11, line 358, change “these” to “many”.
  18. Page 12, line 367 consider removing the word “Surprisingly”—most African American’s descended from slaves abducted from West/Central Africa rather than Northern Africa.
  19. Page 12, line 400 rephrase to read “While the disparity in identified mutations in non-white population versus white populations”…
  20. Page 12, lines 403-407, the preposition “the” is missing in several places.
  21. Page 12, lines 417-418 change the phrase “poorly known recurrent mutations” to ‘in which CFTR variance is understudied or poorly understood.”
  22. Page 12, line 419, consider changing the phrase “in the name of fairness” to “In the interest of health equality”
  23. Page 13, line 44, change “are” to “is”
  24. Page 14, line 473, change “beneficiate” to “benefit”
  25. Page 14, line 478, I believe the authors mean to say “among which are the indications for prenatal testing and implications of results”.
  26. Page 14, line 481 change “to” to “for”.
  27. Page 14, line 491 remove “association” and add “at least one p.Phe508del allele”.

Author Response

Major comment

  1. Please consider grammatical review and editing from someone whose primary language is English (which will help with some of the minor comments below).
    Reply: The English language was reviewed. Problematic formulations were changed.
  2. Page 3, subjects section: For readers to understand the scope of work done at the hospital, consider inclusion of a consort diagram starting with the total number of prenatal ultrasounds done during the period of time.
    Reply: A We thank the reviewer for this advice, we thus added a new figure (Figure 1, page 4) and hope this helps the readers to better understand the strategy.
  3. In the intro, from the author’s previous work, include the sensitivity and specificity of the prenatal ultrasound observed “triad” for diagnosis of CF. Would also consider a table describing the sensitivity and specificity for diagnosis of CF using a single one of the diseases listed in the triad, versus 2 of 3 versus all three. Such a table could help a clinician decide on the utility of suggesting CFTR sequencing for the parents based on both the signs on prenatal ultrasound and other factors (e.g. family history, previous history of miscarriages, etc).
    Reply: The previous work was valid for a population of predominantly Caucasian origin. According to the reviewer advice, we added a table (Table 1, page 2). Although the cohort was quite large, the number of fetuses for each ultrasound pattern remains quite low. Therefore, instead of sensitivity and sensibility, we preferred the use of the likelihood ratio to illustrate the increased risk of CF and reported the numbers of CF fetuses vs non-CF fetuses observed for each pattern in the previous cohort.
  1. Section 3.2 Please reference to which databases the authors are referring (line 204) and be more specific about the in silico tools being utilized to assess pathogenicity (fine to reference legend for figure 2).
    Reply: To clarify, we have changed the order of the figures/tables. The used in silico tools are defined in the “molecular analysis” section (page 2), and in the legend of Table 3 (page 8). Figure 2 becomes Figure 3 and results from a synthesis of Table 2 and Table 3, and thus was placed after Table 3.
  2. The discussion could be shortened as there is some redundancy.
    Reply: In order to gain clarity, we have followed the reviewer’s advice and the discussion has been reorganized and shortened.

Minor comments

  1. Page 2, lines 54-55, please clarify what the phrase “conserved a functional and a clinical variants classification”
    Reply: The phrase has been clarified: “the CF community has conserved two classifications: a functional one devoted to the choice of a genotype-based therapy and a clinical one allowing genetic counseling [4]” (lines 52-54).
  2. Page 2, lines 55-57 it seems that the authors are stating that clinical classification depends on the severity of the genotype dictated phenotype (although there is clearly variability in that correlation as demonstrated by the work done by Mike Knowles in people homozygous for F508del).
    Reply: Classification always leads to simplification. We agree that even F508del homozygous could have variable phenotype. However, they must be considered as severe CF for familial genetic counseling. The phrase was clarified: “Clinical classification depends on the severity of the most frequently associated phenotype when the patients carry a severe mutation in trans.” (line 55)The list that follows is a bit confusing—it needs to be edited, referenced and the terms defined. This remark is very pertinent, and we thank the reviewer. The terms used have been now defined (p2, lines 56-63).
  3. Page 2 line 80, change “are” to “is”
    Reply: revised as requested’ (page 3, line 91)
  4. Define what is meant by the phrase “severe CF mutations” with first use.
    Reply: The term “severe CF mutations” has been replaced by “CF causing mutation”, as the mutations classified as CF causing are considered severe, with very low residual function and associated to the classical form of the disease, as defined line 56.
  5. Page 2, line 86, replace large recruitment with something like “we have the opportunity for diverse recruitment based on the geographic origins of our patient population”. Would also recommend breaking that very long sentence into two sentences. “
    Reply: In our Ile de France laboratory, we have a large recruitment in terms of geographical origins, which is why we have chosen to carry out complete CFTR analysis in both members of the couples in the event of ultrasound suspicion of cystic fibrosis.” Was replaced by “In our Ile de France laboratory, we have the opportunity for diverse recruitment based on the geographic origins of our patient population. Therefore, we have chosen to carry out complete CFTR analysis in both members of the couples in the event of ultrasound suspicion of cystic fibrosis.” (lines 96-100)
  6. Page 2. Line 94: do the authors mean in all patients presenting with bowel abnormalities on prenatal ultrasound, or all patients presenting with signs and symptoms of CF? “
    Reply: all patients” were replaced by “all parents in case of fetal bowel abnormalities” (line 107)
  7. Page 3, line 102, change “retained” to “included”.
    Reply: revised as requested (line 116)
  8. Page 3, lines 102-105, FEB and NVFGB were defined in the intro, so don’t need to be spelled out again here. revised as requested (lines 117-118)
  9. Page 3, lines 118-119: Please explain what is meant by “Depending on the relative emergency of the molecular diagnosis,” e.g. under what circumstances was the diagnosis considered an emergency and why?
    Reply: The method of sequencing was chosen in each case of fetal bowel abnormalities in view to give the results as fast as possible. The word “emergency” was not adapted, since it was not a question of emergency but of quickness. Thus, and the sentence has been modified: “For organizational purpose, sequencing was performed either by Sanger method (primers available on demand), or by high throughput sequencing, without any difference in the quality of the results observed.” (lines 134-137)
  10. Page 4, line 159 change “on” to “of” or “born to”
    Reply: I did not find the word “on” to replace it.
  11. Page 4, line 174: change “satisfying” to “acceptable”.
    Reply: revised as requested (line 196)
  12. Page 4, line 181, change “on” to “of”
    Reply: revised as requested (line 203)
  13. Page 4, lines 185-186, please clarify the points of the sentence starting with CFTR-RD genotype. Do the authors mean “amenable” rather than “eligible”? Neither parent was tested or the pregnancy was multiple?”
    Reply: The text was made more understandable. “CFTR-RD genotype is not eligible to prenatal diagnosis; therefore, both fetuses were not tested” was replaced by “CFTR-RD genotype is not eligible to prenatal diagnosis in France; therefore, the at-risk fetuses were not tested. Thus, we cannot conclude whether the observed phenotype was or was not linked to a CFTR default. “(lines 203-204)
  14. Page 11, lines 320-321, do the authors mean other fetuses reported in the literature or in their previous case study or? “
    Reply: This triad of signs is very rare, since it was observed only in the CF fetus in our series, and not in other fetuses” was replaced by “This triad of signs is very rare in our experience: in our previous study, we observed this triad in 8 fetuses (1.15%) among the 694 cases referred for ultrasound digestive abnormalities [16], and in the present study we observed it only once among 132 fetuses.” (page 11, lines 333-336)
  15. Page 11 line 345-add references after abnormalities.
    Reply: References have been added line 355
  16. Page 11, line 355, add “the” before Asian.
    Reply: revised as requested (line 364)
  17. Page 11, line 358, change “these” to “many”.
    Reply: revised as requested (line 367)
  18. Page 12, line 367 consider removing the word “Surprisingly”—most African American’s descended from slaves abducted from West/Central Africa rather than Northern Africa.
    Reply: revised as requested (line 380), the text has been reworked.
  19. Page 12, line 400 rephrase to read “While the disparity in identified mutations in non-white population versus white populations”… “
    Reply: While the disparity of mutations with respect to the concerned population has been known for a long time,” has been replaced by “While the disparity in identified mutations in non-Caucasian population versus Caucasian populations has been known for a long time,” (page 12, lines 408-409)
  20. Page 12, lines 403-407, the preposition
    Reply: “the” is missing in several places. “the” was added in several places (page 12, lines 407-417)
  21. Page 12, lines 417-418 change the phrase “poorly known recurrent mutations” to ‘in which CFTR variance is understudied or poorly understood.”
    Reply: revised as requested (page 12, line 423)
  22. Page 12, line 419, consider changing the phrase “in the name of fairness” to “In the interest of health equality”
    Reply: revised as requested (page 12, line 424)
  23. Page 13, line 44, change “are” to “is”
    Reply: revised as requested (page 13, line 431)
  24. Page 14, line 473, change “beneficiate” to “benefit”
    Reply: revised as requested (page 14, line 461)
  25. Page 14, line 478, I believe the authors mean to say “among which are the indications for prenatal testing and implications of results”.
    Reply: You are wright. . “among which the indication of prenatal diagnosis » was replaced by « among which are the indications for prenatal testing and implications of the results” (line 464-465)
  26. Page 14, line 481 change “to” to “for”.
    Reply: revised as requested (line 468)
  27. Page 14, line 491 remove “association” and add “at least one p.Phe508del allele”.
    Reply:revised as requested (line 478)

Reviewer 2 Report

This paper describes genetic testing of the parents of fetuses detected on antenatal ultrasound as having digestive tract features associated with cystic fibrosis (CF). The abnormalities include hyperechogenic bowel, failure to detect the gall bladder and dilated bowel loops. Of 208 fetuses, only one was diagnosed with CF, and this one was the only fetus to present with all 3 of the manifestations above.

The paper then goes on to describe the gene variants (some CF-causing, others less definitively so) found in the parents, who were of very mixed origins and ethnic backgrounds. It also describes the role of geneticists and the genetics lab in this process before concluding that an extensive search for CFTR gene variants is warranted in all cases presenting with a fetus with any of the above abnormalities.

My overarching consideration of the paper is that it is too long and does not clearly enough set out its aims. It reads more like a thesis, with multiple strands, several of which are presented in enormous detail. My specific comments are:

  1. Throughout, I found the authors categorising VVCC and CFTR-RD separately as very confusing. Most CF teams, and indeed the cftr2 resource, would consider them either the same or at the very least greatly overlapping. VVCC can lead in some people to apparent good health (many picked up on NBS and terms CF-SPID in Europe/ CRMS in US), in others to CFTR-related disorders, and only rarely to a more classical, albeit usually pancreatic sufficient form of CF itself. This needs better explaining and justifying if the authors wish to retain this separation.
  2. I found the section describing the antenatal US abnormalities and their overlap useful and informative. What was lacking though was any detail around the cases which were not picked up as CF, ie all but 1. Did these fetuses have another condition diagnosed, or the infants after birth? Two cases of variants associated with CFTR-RD were mentioned, and on line 185: ‘CFTR-RD genotype is not eligible to prenatal diagnosis; therefore, both fetuses were not tested’. What about their post-natal course though? Did they have any testing beyond the newborn screen for CF? Even if they were NBS negative, so was the one case with CF; had they inherited both parental variant alleles? Did they undergo sweat testing? We’re left not being able to interpret the genetic data in these cases and understand whether it was relevant.
  3. Line 189 on: the description of a mother with two variants is off topic and could easily be removed.
  4. I found the section presenting gene variants in people from the different regions excessively long; as these data only arose from the fetal abnormalities, we cannot conclude anything about the allele frequencies in these populations. The table is good and a much better way to describe the findings.
  5. Overall, the conclusions seem somewhat overstated:
    1. Line 91 ‘We confirm the 91 high risk of CF in the case of the triad of digestive signs….’, yes 100% of cases, but only n=1
  6. The title: ‘…search for frequent mutations is not enough’, ditto, for that one case. Hard to argue the case for any other families as no clinical relevance is placed around the results arising on more extensive testing.

Reviewer 3 Report

The authors present the results of the in depth molecular analysis of the CFTR gene in the context of ultrasound suspicion of cystic fibrosis. They  gathered a substatial number of fetuses (n=108) and gave detailed clinical and ethnic data.

This study is of particular interest as it allowed to obtain genetics data in fetuses of different ethnical origins, that would not be  available with the search of frequent CF mutations only (recommended as a first-line strategy). This could be realy useful to review recommendations for genetics testing in this indication. However, I have some comments.

1/ Major comments :

The authors should not be so affirmative about the interest of in depth CFTR analysis in the context of ultrasound suspicion of CF :

  • the only CF fetus of the cohort is, certainly, homozygous for a rare variant but in a context of consanguinity. Thus, the fetus would have been diagnosed as in depth analysis is recommended if the parents share a common haplotype.
  • almost half of the mutations detected are CFTR-RD variants, including variants with low penetrance : very low risk for CF
  • a third of variants are of unknown or unclrear significance, for which genetic counselling is challenging.
  • Except for the CF fetus, no pathogenic variant was found in the partners of the parents carrying the CF/VCC/CFTR-RD/VUS.
  • In the previous study of 694 fetuses (deBecdelièvre et al., 2010), considered as a reference, only two CF fetuses were carriers of 2 rare CF mutations, that represents 0.2% of cases.

Thus, I would propose to the authors:

  • to modify the title as follows : "Prenatal ultrasound suspicion of cystic fibrosis in a multiethnic population: is the search for frequent mutations enough ?
  • to shorten the discussion (the same message is found in several subsections) and to discuss in more details the interest or the difficulty to identify CFTR-RD and VUS : what can we propose to the couples and in term of follow up to the children once born? Is there an interest to identify such variants if nothing can be proposed afterwards ?

2/ Minor comments:

English could be improved

Line 202: please check the number of parents carrying a VCC variant (I counted only 5 parents)

Table 2: 

  • classification of c.3718-2477C>T in CFTR2 is CF(-causing) but I did not see the note about pancreatic status : if you proposed this classification, according to the data available in CFTR2, it shoud be specified.
  • Column "CFTR-France" for c.3718-2477C>T/Q237E: replace "/" by ";" to be consistent with other.
  • Replace "unknown" by "NA" in the column "protein name" and by "not reported" or "not found" (to avoid confusion with unknwn significance) in the columns "classification in CFTR2/CFTR-France
  • Line "TG12T5/TG11T5/V562I" : add all classes in the column CFTR-France: CFTR-RD;VUS;VUS

Round 2

Reviewer 3 Report

I'd like to thank the authors for taking my comments and suggestions in consideration. The modifications of the manuscript meet my requirements. Indeed, even if I'm not fully convinced by a systematic coprehensive CFTR study, particularly due to the detection of VUS that are difficult to manage in this context, the modifications allow to consider the results in a more open-minded and critical view.

Finally, the shorter discussion sections appears clearer and raises the key points of the results and of international literature.

I have few minor comments on the text :

  • line 110 (page 3/19): We describe the CF genotype/phenotype in a fetus... --> I'd precise "genotype/phenotype relationship" or "G/P correlation"
  • table 2 (pages 6 and 7/19): in the column dbSNP, replace "unknown" by "no rs" (typo error in the dbSNP column of C1410X: remove the "T")
  • table 2 (page 7/19): in last line of the "Northern Africa" part of the table (line L997F/TG12T5/S977F) should "unknown" be replaced by "not reported" like for the L997F alone ?
  • Figure 3 (page 9/19): change "VCC" by "VVCC" on teh graph to be consistent with the legend
  • Line 330 (page 12/19) : I would rephrase "In the 20 individuals originating from Asia, only one had a rare variant, the TG13T5 allele,  at the heterozygous state."

Author Response

We thank the reviewer for his wise advice and agree that the interpretation of comprehensive CFTR studies raises complex questions.

Please find the answers to the comments bellow :

  • line 110 (page 3/19): We describe the CF genotype/phenotype in a fetus... --> I'd precise "genotype/phenotype relationship" or "G/P correlation" . This was precised : “We describe the genotype/phenotype relationship in a CF fetus with….”
  • table 2 (pages 6 and 7/19): in the column dbSNP, replace "unknown" by "no rs" (typo error in the dbSNP column of C1410X: remove the "T") This was changed. Thank you for this improvement.
  • table 2 (page 7/19): in last line of the "Northern Africa" part of the table (line L997F/TG12T5/S977F) should "unknown" be replaced by "not reported" like for the L997F alone ? Yes. The replacement was done.
  • Figure 3 (page 9/19): change "VCC" by "VVCC" on teh graph to be consistent with the legend. This was changed.
  • Line 330 (page 12/19) : I would rephrase "In the 20 individuals originating from Asia, only one had a rare variant, the TG13T5 allele,  at the heterozygous state." The sentence has been changed, as recommended.

We thank the reviewer for his contructive reading of our work.